

# Comparing quantile regression spline analyses and supervised machine learning for environmental quality assessment at coastal marine aquaculture installations

Kleopatra Leontidou*, Verena Rubel* and Thorsten Stoeck

Ecology Group, Rheinland-Pfälzische Technische Universität Kaiserslautern-Landau, Kaiserslautern, Germany
* These authors contributed equally to this work.

Corresponding author
Thorsten Stoeck,
stoeck@rhrk.uni-kl.de

## ABSTRACT

Organic enrichment associated with marine finfish aquaculture is a local stressor of marine coastal ecosystems. To maintain ecosystem services, the implementation of biomonitoring programs focusing on benthic diversity is required. Traditionally, impact-indices are determined by extracting and identifying benthic macroinvertebrates from samples. However, this is a time-consuming and expensive method with low upscaling potential. A more rapid, inexpensive, and robust method to infer the environmental quality of marine environments is eDNA metabarcoding of bacterial communities. To infer the environmental quality of coastal habitats from metabarcoding data, two taxonomy-free approaches have been successfully applied for different geographical regions and monitoring goals, namely quantile regression splines (QRS) and supervised machine learning (SML). However, their comparative performance remains untested for monitoring the impact of organic enrichment introduced by aquaculture on marine coastal environments. We compared the performance of QRS and SML using bacterial metabarcoding data to infer the environmental quality of 230 aquaculture samples collected from seven farms in Norway and seven farms in Scotland along an organic enrichment gradient. As a measure of environmental quality, we used the Infaunal Quality Index (IQI) calculated from benthic macrofauna data (reference index). The QRS analysis plotted the abundance of amplicon sequence variants (ASVs) as a function to the IQI from which the ASVs with a defined abundance peak were assigned to eco-groups and a molecular IQI was subsequently calculated. In contrast, the SML approach built a random forest model to directly predict the macrofauna-based IQI. Our results show that both QRS and SML perform well in inferring the environmental quality with 89% and 90% accuracy, respectively. For both geographic regions, there was high correspondence between the reference IQI and both the inferred molecular IQIs ($p < 0.001$), with the SML model showing a higher coefficient of determination compared to QRS. Among the 20 most important ASVs identified by the SML approach, 15 were congruent with the good quality spline ASV indicators identified *via* QRS for both Norwegian and Scottish salmon farms. More research on the response of the ASVs to organic enrichment and the co-influence of other environmental parameters is necessary to eventually select the most powerful stressor-specific indicators. Even though both approaches are promising to infer

environmental quality based on metabarcoding data, SML showed to be more powerful in handling the natural variability. For the improvement of the SML model, addition of new samples is still required, as background noise introduced by high spatio-temporal variability can be reduced. Overall, we recommend the development of a powerful SML approach that will be onwards applied for monitoring the impact of aquaculture on marine ecosystems based on eDNA metabarcoding data.

## INTRODUCTION

Marine coastal finfish aquaculture is a growing industry to satisfy the global seafood demand while relieving pressure on wild fish stocks through fisheries (*FAO, 2020*). The deposition of uneaten feed and fish feces at aquaculture installation sites can lead to disturbance of the local benthic ecosystem (*Carroll et al., 2003*). The accumulation of this nitrogen-rich organic material on the seafloor stimulates the activity of benthic bacterial communities and the breakdown of organic material may eventually lead to oxygen depletion and changes of benthic community structures and functions (*Bissett et al., 2007*; *Fodelianakis et al., 2015*; *Karakassis & Hatziyanni, 2000*). To maintain ecosystem services, it is crucial that ecosystem function(ing) does not alter beyond repair. This requires a frequent monitoring of aquaculture installation sites, which traditionally relies on the inference of a biological index using benthic macroinvertebrate bioindicators, which is then translated into an environmental quality status. Such an index is, for example, the Infaunal Quality Index (IQI). As part of the Water Framework Directive (WFD; 2000/60/EC) the IQI provides an ecological status assessment of marine environments based on the inventory of benthic macroinvertebrate communities. Three metrics contribute to the IQI: the number of taxa, Simpson's Evenness and the AZTI Marine Benthic Index (AMBI; *Borja, Franco & Pérez, 2000*), with the first two indicating the level of biodiversity and the third being a measure of response to anthropogenic disturbance (*Phillips et al., 2014*). IQI ranges from zero to one, with values close to one indicating a "high" environmental quality of unimpacted sites and values close to zero a "low" environmental quality, while the "good/moderate" boundary is defined at 0.64 (*Phillips et al., 2014*).

Creating inventories of benthic macroinvertebrate communities is extremely time consuming, expensive, and with low potential for upscaling in high-throughput monitoring (*Pawlowski, Apothéloz-Perret-Gentil & Altermatt, 2020*). Therefore, the interrogation of benthic bacterial communities using eDNA-metabarcoding and the inference of environmental quality (EQ) status from bacterial DNA sequences represented by amplicon sequence variants (ASVs) has emerged as a very powerful alternative to the traditional microscopy-based macroinvertebrate biomonitoring (*Aylagas et al., 2017*; *Birrer et al., 2018*). Bacteria are currently targeted as bioindicators in environmental monitoring, since they react faster than macroinvertebrates to environmental changes (*Keeley, Wood & Pochon, 2018*; *Lanzén et al., 2021*) due to their shorter generation times

(*Lear et al., 2011*; *Nogales et al., 2011*). The ASV approach infers unique biological sequences directly from HTS sequencing data, which can be used for downstream analysis (*Callahan, McMurdie & Holmes, 2017*).

A major challenge in the exploitation of bacterial ASVs as bioindicators is the translation of their abundance-distribution patterns into EQ categories. Two different approaches have been successfully established. One approach uses supervised machine learning (SML) to predict EQ from the bacterial ASV composition found at a specific site (*e.g.*, *Cordier et al., 2018*; *Armstrong & Verhoeven, 2020*; *Dully et al., 2021*). Machine learning aims to develop computer algorithms that can "learn" from a set of data and improve their performance with experience, to assist in big data and complex classification tasks (*Cordier et al., 2021*). The strategy of SML is to train a predictive model using a labeled dataset, of which the solution (*e.g.*, environmental quality status) is known for each provided sample, in order to classify upcoming samples without labels (*e.g.*, unknown environmental quality status). The training of such a model consists in identifying, among features (such as bacterial ASVs), the ones (or a combination of them) that correlate or explain the known solutions of the training observations. This extracted knowledge is then used by the predictive model (algorithm), trained on only a subset of the possible real-world situations, to make predictions on upcoming samples of unknown environmental quality. Advantages of the SML approach are that the algorithms are best fitted for large and noisy datasets, including the analysis of highly dimensional microbial genomics datasets (*Knights, Costello & Knight, 2011*; *Beck & Foster, 2014*; *Smith et al., 2015*). Furthermore, this approach is computationally fast and requires relatively little resources (*Breiman, 2001*), and the ecological signal of features (individual ASVs) and association rules within the full (bacterial ASV) dataset are automatically disentangled from background noise (*Prasad, Iverson & Liaw, 2006*; *Fox et al., 2017*). One further decisive strength of the SML approach is that it does not rely on taxonomic and ecological information of the detected bacterial ASVs in a dataset, and, thus, is not sensitive to gaps in nucleic acid reference databases (*Cordier et al., 2018*). Furthermore, an underlying statistical framework allows assessment of model prediction accuracy (*Landis & Koch, 1977a*) and SML is easily up-scalable and fully automatable (*Cordier et al., 2018*). However, a sufficient amount of training data is required for accurate predictions of new uncharacterized samples (*Lanzén et al., 2021*; *Breiman, 2001*). For the successful implementation of SML, the number of samples needed for the solution of a specific problem (here: classification of environmental quality at salmon aquaculture installations) is under investigation (*Dully et al., 2021a*), while the sample coverage across environmental gradients is also being discussed (*Lanzén et al., 2021*).

An alternative approach, successfully applied for the inference of environmental quality in marine environments, relies on the *de novo* identification of bacterial indicator ASVs *via* quantile regression spline analysis (QRS). The principle of this approach is to statistically identify the abundance peak of an organism or ASV along an environmental gradient. Each organism or ASV with a defined peak-abundance along a specific environmental gradient can then be classified as a specific biomarker for the value range of the corresponding environmental parameter, and, thus, be allocated to a specific eco-group.

An eco-group includes all species that show a similar abundance pattern along a gradient. For example, benthic invertebrates dominating sites with organic enrichment pollution would be categorized in an eco-group of opportunistic species, while the ones that become abundant in undisturbed environments would be categorized in an eco-group of sensitive species (*Grall & Glémarec, 1997*). Based on indicator ASVs and their eco-group assignments, a molecular biotic index can be calculated to infer an EQ classification of the samples under survey (*Keeley, Wood & Pochon, 2018*; *Lanzén et al., 2021*; *Aylagas et al., 2021*). Like the SML approach, QRS analysis is a taxonomy-independent *de novo* approach for the identification of bioindicators, which bypasses limited taxonomic and ecological information that is available for most bacterial ASVs (*Cordier et al., 2021*). This means that more diversity can be exploited for monitoring purposes which can help us get different perspectives on an ecosystem's state. Computationally, also this approach is easily up-scalable and fully automatable. A possible weakness of the QRS approach may be to cope with noisy datasets. QRS bioindicator inference requires a highly consistent response of the abundance of a bacterial ASV along an environmental gradient such as organic enrichment. However, several seasonal and local environmental effects may co-influence bacterial abundance patterns in addition to the environmental pressure variable of interest. This may compromise the identification of bioindicators *via* QRS whereas SML algorithms could be trained by incorporating co-variables (*Frühe et al., 2021a*).

While both methods were put to the test for the same coastal samples subjected to urban discharge (*Lanzén et al., 2021*), the performance of SML *vs* QRS-inferred bioindicators has thus far gone untested for coastal aquaculture installations. Towards the development of a standard operating procedure (SOP) for compliance monitoring of aquaculture effects on marine coastal environments, we here for the first time compared the performance of both approaches for 230 samples from Norwegian and Scottish salmon aquaculture installations. The obtained results were then compared to ground truth results obtained from traditional compliance monitoring of the same sampling sites using benthic macroinvertebrate surveys.

## MATERIALS AND METHODS

### Sampling and data acquisition

Data for this study consist of two parts. The first part includes previously published metabarcode data (V3–V4 region of the SSU rRNA gene) from benthic samples of Atlantic salmon (*Salmo salar*) aquafarm installations. These data include Illumina amplicon sequences obtained from 138 sediment samples of seven Norwegian salmon farms collected during compliance monitoring (*Frühe et al., 2021b*), available at the Sequence Read Archive (SRA) of National Centre for Biotechnology Information (NCBI) under BioProject number PRJNA562304, and Illumina amplicon sequences from 18 sediment samples of two Scottish salmon farms (S03, S04) obtained from our previous studies (*Dully et al., 2021b*; *Frühe et al., 2021b*), which are available under SRA BioProject number PRJNA768445 (S03) and PRJNA666305 (S04), respectively.

The second part of the dataset analyzed in this study consists of new metabarcode data, which we obtained from 74 sediment samples of five further Scottish salmon farms (S01,

S02, S05, S06, S07) collected during compliance monitoring of these farms. Sampling occurred in the same way as described for the previously published data that we used as part of this study (see above, *Dully et al., 2021b* and *Frühe et al., 2021b*). In brief, sediment was collected at 3–10 stations (depending on farms, see Table S1) along an organic enrichment transect extending from cage edges (CE) to reference sites (REF) at least 500 m distant from the aquafarm installations in the direction of the prevailing current flow. At each site, two biological replicates were taken from a van Veen grab (0.1 m$^2$ area), each replicate consisting of ca. 20 g of surface sediment (upper few millimeters) collected using plastic spatulas. Immediately after collection, samples were preserved in LifeGuard solution (Qiagen, Hildesheim, Germany) (equal volume buffer to sediment) and frozen at −20 °C upon arrival in the laboratory until further processing.

The remaining sediments of the van Veen grabs were washed through a 1-mm sieve and the residue was fixed in 4% borax-buffered formaldehyde to collect benthic macroinvertebrates for microscopic macrofauna analysis (compliance monitoring).

## Assigning samples to environmental quality categories

Benthic macrofaunal species lists obtained during compliance monitoring of the salmon farms under study were provided by the companies operating these salmon farms. From these macrofauna matrices we then calculated the Infaunal Quality Index (IQI) according to *Phillips et al. (2014)*, with inference of the AMBI using AZTI's AMBI tool (https://ambi.azti.es). IQI values for the previously published samples used in this study (*Dully et al., 2021b*; *Frühe et al., 2021b*) were obtained in the same way. According to the IQI "good/moderate" decision boundary (*Scottish Environmental Protection Agency (SEPA), 2018*), all samples were then assigned either to the IQI category ≥0.64 (very good to good environmental quality) or <0.64 (moderate to poor environmental quality). These data were then used as reference (ground truth) for the downstream statistical comparisons with the EQ classification of the same samples that we obtained from bacterial ASVs *via* SML and QRS (see below). The IQI classes of our samples based on IQI intervals defined in *Phillips et al. (2014)* and translated as environmental status of the samples are shown in Fig. S1.

## DNA extraction, PCR amplification and sequencing

Bacterial sequence amplicons were obtained as described in detail in *Frühe et al. (2021b)*. In brief, total eDNA was extracted from homogenized sediment (ca. 250 mg) using the DNeasy PowerSoil kit (Qiagen) following the manufacturer's protocol. The hypervariable V3–V4 region of the SSU rRNA gene (ca. 450 bp) was amplified using the primer pair Bakt_341F (CCTACGGGNGGCWGCAG) and Bakt_805R (GACTACHVGGGTATCTAATCC) (*Herlemann et al., 2011*). Three technical replicate PCR reactions were performed for each sample to minimize potential PCR bias. The cycling conditions employed an initial activation step of NEB's Phusion High-Fidelity DNA polymerase (NEB, Ipswich, MA, USA) at 98 °C for 30 s and 27 cycles consisting of 98 °C for 10 s, 62 °C for 30 s, and 72 °C for 30 s, followed by a final 5-min extension at 72 °C. The quality of the resulting PCR products was checked on an 0.8% agarose gel.

The three replicates of the same sample were pooled prior to purification with the MinElute PCR purification kit (Qiagen, Hilden, Germany). Sequencing libraries were constructed using the NEB Next Ultra™ DNA Library Prep Kit for Illumina including a standard negative control of a DNA template-free library. The quality of the libraries was checked with an Agilent Bioanalyzer 2100 system (Agilent Technologies, Santa Clara, CA, USA). Libraries were then sequenced on an Illumina MiSeq platform (Illumina, San Diego, CA, USA), generating 2 × 250 bp paired end reads at SeqIT GmbH & Co.KG (Kaiserslautern, Germany). Raw sequences are deposited under SRA BioProject number PRJNA947566.

## Bioinformatics analysis and data preparation

Raw bacterial V3–V4 sequence reads that were newly produced in this study (five Scottish salmon farms, 74 samples) and the original V3–V4 datasets obtained from GenBank (two Scottish salmon farms, consisting of 18 samples plus seven Norwegian salmon farms consisting of 138 samples) were quality filtered and trimmed using the dada2 pipeline (*Callahan et al., 2016*) in R Studio 3.5.1, as described in *Dully et al. (2021)*. Truncation length was set to 230 bp so that the phred quality score reaches >30 for at least 51% of all reads corresponding to 99.9% base call accuracy (*Ewing et al., 1998*). For maxEE we chose one to maximize downstream sequence quality. The paired-end sequences were merged using minimum 20 bp overlap and a mismatch of two bases was allowed. Amplicon Sequence Variants (ASVs) were inferred based on an error rate model that we generated for each sequence run independently, which removes errors introduced during PCR amplification and sequencing (*e.g.*, base substitution errors). The sequences were checked for chimeras using the *uchime_denovo* function of *vsearch* (*Rognes et al., 2016*). Taxonomic assignment was conducted using *vsearch's syntax* function based on the Greengenes database (*McDonald et al., 2012*) and the last common ancestor approach. To analyze sequencing depth, saturation curves for each dataset were constructed using the *rarecurve* function of the *vegan* package (*Oksanen et al., 2020*).

Finally, we produced two individual ASV-to-sample matrices, one of which included all samples from seven Norwegian salmon farms (*n* = 138) and one including all samples (*n* = 92) from seven Scottish salmon farms. These ASV-to-sample-matrices were then converted to relative abundance tables representing the relative proportions (in percent) that each ASV contributed to an individual sample using the function "*prop.table*" in R to compensate for any differences in the sequencing depth among samples (*Aylagas et al., 2016*; *Gerhard & Gunsch, 2019*; *Dully et al., 2021a*). The mean relative abundance of each ASV was then calculated across all samples per ASV-to-sample matrix and, following the example of *Keeley, Wood & Pochon (2018)*, we then chose the 250 ASVs with the highest mean relative abundance (contributing ≥0.04% to the total number of reads) in each of the two datasets for the molecular IQI inference.
## Quantile regression splines (QRS) analyses, eco-group assignment and molecular IQI inference (*mol*-IQI$_{QRS}$)

To identify bacterial indicators across an environmental quality gradient (here IQI) using QRS analyses we followed the workflow described in *Keeley, Wood & Pochon (2018)*.

In the first step, QRS models for the 95th percentile were constructed for the 250 most abundant ASVs in each of the two ASV-to-sample matrices, using the R packages "quantreg" (version 5.86; *Koenker, 2021*) and "splines" (*R Core Team, 2020*). Per salmon farm, the relative ASV read abundance of each sample (response variable) was plotted against the macrofauna-obtained IQI values of these same samples (predictor variable). A total of 1,750 regression spline models were generated for each, the Norwegian and the Scottish salmon farm datasets (in each case: 250 most abundant ASVs × seven salmon farms). We then used the "*find_peaks*" function of the R package "pracma" (*Borchers, 2021*) to identify the IQI values at which each ASV peaked in its abundance.

In the next step, each ASV had to be assigned to an eco-group based on its QRS-inferred IQI peak data. Therefore, out of the 250 analyzed ASVs in each dataset, we identified ASVs that had good quality splines. The general criterion for a good quality spline is a consistent response of an ASV across a dataset (*Keeley, Wood & Pochon, 2018*; *Aylagas et al., 2021*). We here defined a consistent response as follows: A good quality spline applied if an ASV was characterized by an IQI peak in at least five out of seven farms per dataset (conservative majority rule) and a standard deviation (SD) of <0.2 of the mean IQI-peak across all seven farms within a dataset. Our reasoning for choosing an SD of <0.2 is based on the translation of IQI values into eco-groups according to *Phillips et al. (2014)*: IQI 0–0.24 = Eco-Group V, IQI 0.25–0.43 = Eco-Group IV, IQI 0.44–0.63 = Eco-Group III, IQI 0.64–0.74 = Eco-Group II, IQI 0.75–1 = Eco-Group I, with Eco-Group I corresponding to very sensitive taxa and Eco-Group V to opportunistic species. Thus, a value of 0.2 matches the averaged IQI-interval of an individual eco-group and an SD exceeding this interval may encompass two different Eco-Groups.

For downstream analyses we then only selected ASVs with good quality splines as potential bioindicators for the calculation of the QRS-inferred molecular IQI (*mol*-IQI$_{QRS}$).

Using the Eco-Group assignments of each good quality spline ASV according to *Phillips et al. (2014)* (see above), we then calculated the molecular AMBI index (*Borja, Franco & Pérez, 2000*) as follows.

$$Mol-AMBI = ([1.5 \times \%Eco-Group\,II] + [3 \times \%Eco-Group\,III] \\ + [5 \times \%Eco-Group\,IV)] + [6 \times \%Eco-Group\,V])/100 \tag{1}$$

Finally, a QRS-based *mol*-IQI$_{QRS}$ was inferred using the IQI version IV by the Water Framework Directive (*European Parliament & Council, 2000*; *Phillips et al., 2014*) as follows:

$$Mol-IQI = ([0.38 \times ((1 - (mol-AMBI/7))/(1 - (mol-AMBI_{Ref}/7)] \\ + [0.08 \times ((1 - \lambda')/(1 - \lambda')_{Ref})] + [0.54 \times (S/S_{Ref})^{0.1}] - 0.4)/0.6 \tag{2}$$

Where: AMBI corresponds to the *mol*-AMBI as calculated by Eq. (1); $1 - \lambda'$ is Simpson's Evenness index; S is the $\log_{10}$ number of ASVs; Ref corresponds to reference values of unimpacted sites. The diversity metrics were calculated using the R package "vegan".

## Supervised machine learning and molecular IQI inference (*mol*-IQI$_{RF}$)

Data input for the random forest (RF) regression analysis were the two relative abundance ASV-to-sample matrices of the Scottish and Norwegian salmon farms (features) and the macrofauna-IQI values (reference labels). The 250 most abundant ASVs in each of the two sub-datasets (Scottish and Norwegian) were selected for RF-based IQI inference, as selected for the QRS analysis. RF was conducted using the R package "*randomForest*" for classification and regression (v. 4.6.14, *Liaw & Wiener, 2002*) with default parameter settings for RF regression (Number of variables tried at each split *mtry* = No. of variables/3; Number of trees *ntree* = 500).

All samples were subjected to a leave-one-out cross validation (LOOCV) run (*James et al., 2013*). In each run, one single observation (=one sample) was omitted from the ASV table to build a regression model with all the remaining samples (=137 samples in case of Norwegian farms and 91 samples in case of Scottish farms). The previously omitted observation (sample) was then used to validate the regression model built on the remaining observations (samples). Because each sample was used once for validation, this resulted in 138 independent models for the Norwegian sub-dataset and 92 independent models for the Scottish sub-dataset. The LOOCV approach was performed 10 times for each sample individually resulting in 1,380 models for the Norwegian dataset and 920 models for the Scottish dataset. Regression prediction of IQI values was averaged from all constructed RF regression models for the Scottish and Norwegian salmon farms dataset. The RF variable importance measure was also determined across all respective models by calculating the average value.

In accordance with the macrofaunal-inferred EQ reference values (see above), the RF-predicted *mol*-IQI$_{RF}$ samples were subsequently grouped into the two IQI categories.

## Assessing accuracies of *mol*-IQI$_{QRS}$- and *mol*-IQI$_{RF}$-derived environmental quality (EQ) classifications

All individual samples from the Scottish and the Norwegian salmon farms obtained three IQI values as a measure of environmental quality index. The first IQI value (IQI$_{MA}$) was obtained from macrofauna species surveys (compliance monitoring of the salmon farms under study). This IQI is the traditional reference IQI that we consider as the statistical ground truth. The second IQI value (*mol*-IQI$_{QRS}$) was obtained from *de novo* identification and characterization of bacterial ASV bioindicators *via* quantile regression spline analyses. The third IQI value (*mol*-IQI$_{RF}$) resulted from random forest predictions. All samples were then classified according to the IQI 0.64-good/moderate boundary (see above) based on their *mol*-IQI$_{QRS}$ and on their *mol*-IQI$_{RF}$ values. We then inferred the accuracy of the *mol*-IQI$_{QRS}$ classification and *mol*-IQI$_{RF}$ classifications by calculating the relative number of correct pairwise predictions compared to the IQI$_{MA}$ reference classifications. To further

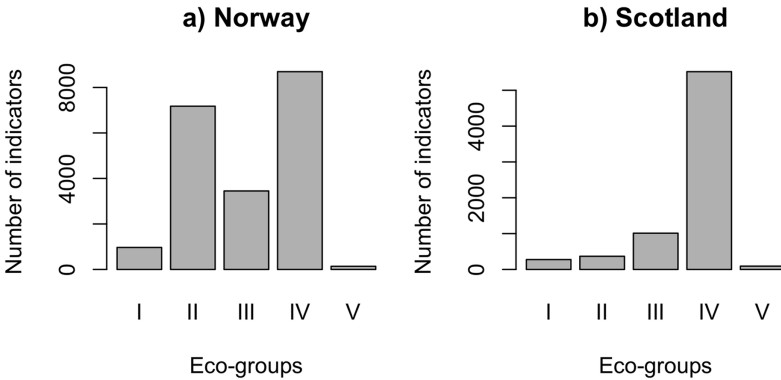

**Figure 1** Number of indicators assigned to each eco-group in (A) Norway (*n* = 138 samples) and (B) Scotland (*n* = 92 samples), with Eco-Group I corresponding to very sensitive taxa and Eco-Group V to opportunistic ones.

examine the relationship between $IQI_{MA}$ and both *mol*-IQIs, regression analyses were performed using the *lm* function of the R package. The highest coefficient of determination ($R^2$) from these analyses in combination with the significance level identified the approach (QRS or RF) with the best predictive power for EQ classification (above or below good/moderate boundary). Finally, the agreement between the reference $IQI_{MA}$ and the *mol*-$IQI_{QRS}$ as well as between the reference $IQI_{MA}$ and the *mol*-$IQI_{RF}$ was tested with Cohen's kappa statistics using the *kappa2* function (squared weight) of the R package "irr" (v 0.84.1, *Gamer et al., 2012*). Kappa values above 0.8 indicate "almost perfect agreement" (*Landis & Koch, 1977b*).

## RESULTS

### Data overview

The raw data of the seven Norwegian salmon farms installations consisted of 22,029,762 raw reads in total which were bioinformatically filtered for High-Quality (HQ) sequences, resulting in 3,541,124 HQ reads (grouped in 66,085 ASVs).

For Scotland, seven farms in total have been investigated, from which we obtained 20,992,048 raw reads. After DADA2 processing, we were able to retain 5,229,185 HQ reads (grouped in 79,511 ASVs). For each farm, a detailed sequence overview per sample is provided in Table S1 and their rarefaction curves are provided in Fig. S2.

The 250 most abundant ASVs used for the QRS and RF analyses corresponded to 1,536,500 reads for Norway (accounting for 43% of the total dataset) and 2,206,423 reads for Scotland (accounting for 42% of the total dataset).

### Eco-group assignment and identification of potential bacterial ASV indicators

QRS analysis resulted in 148 indicator ASVs with good quality splines for the Norwegian salmon farm, corresponding to 59% of the top 250 most abundant ASVs. For the Scottish salmon farm dataset, we identified only 79 indicator ASVs with good quality splines, corresponding to 32% of the top ASVs. Good quality splines are shown in Fig. S3 and

a) Norway

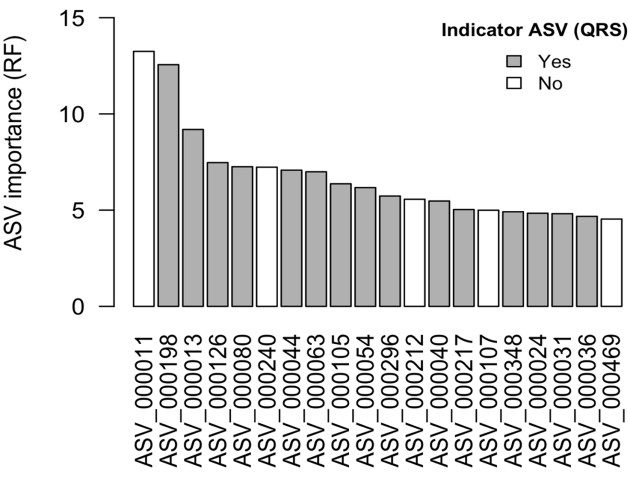

b) Scotland

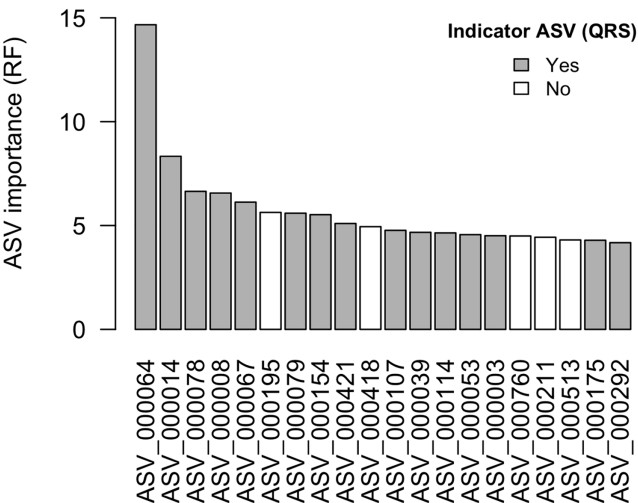

**Figure 2 Top 20 ASVs with the highest importance value which was assigned by random forest (RF) for (A) Norway and (B) Scotland.** Indicated with grey are the ASVs which were identified as indicators with quantile regression splines (QRS).

summarized in Table S2. The majority of the ASVs from Norwegian salmon farms with good quality splines could be assigned to Eco-Groups II ($n$ = 52), III ($n$ = 25) and IV ($n$ = 63) (Fig. 1A). Good quality spline ASVs of sensitive (Eco-Group I) and opportunistic (Eco-Group V) bacteria were scarce ($n$ = 7 and 1, respectively). In case of Scottish salmon farm samples, the vast majority of good quality spline ASVs belonged to Eco-Group IV ($n$ = 60, corresponding to 76% of all good quality spline ASVs). In comparison to Eco-Group IV, good quality spline ASVs are underrepresented for all other eco-groups in Scottish salmon farm samples (Fig. 1B).

**Table 1 ASVs assigned with the highest variable importance by random forest (RF) and simultaneously identified as indicators *via* quantile regression splines (QRS) for (A) Norway and (B) Scotland.**

| ASV | Eco-Group | Taxa name | Variable importance |
|---|---|---|---|
| (A) Norway | | | |
| ASV_000198 | II | Gammaproteobacteria | 12.56 |
| ASV_000013 | IV | Helicobacteraceae | 9.19 |
| ASV_000126 | II | *Nitrospina* | 7.47 |
| ASV_000080 | IV | *Desulfosarcina* | 7.26 |
| ASV_000044 | IV | Flavobacteriaceae | 7.08 |
| ASV_000063 | II | Rhodospirillales | 6.99 |
| ASV_000105 | IV | Helicobacteraceae | 6.37 |
| ASV_000054 | II | Syntrophobacteraceae | 6.17 |
| ASV_000296 | IV | Helicobacteraceae | 5.73 |
| ASV_000040 | II | Myxococcales | 5.47 |
| ASV_000217 | III | Acidobacteria | 5.03 |
| ASV_000348 | IV | Bacteroidales | 4.91 |
| ASV_000024 | IV | Alteromonadales | 4.84 |
| ASV_000031 | III | Myxococcales | 4.81 |
| ASV_000036 | IV | Bacteroidales | 4.68 |
| (B) Scotland | | | |
| ASV_000064 | IV | *Lutimonas* | 14.67 |
| ASV_000014 | IV | Helicobacteraceae | 8.33 |
| ASV_000078 | IV | Helicobacteraceae | 6.65 |
| ASV_000008 | IV | *Psychrilyobacter* | 6.56 |
| ASV_000067 | IV | Helicobacteraceae | 6.13 |
| ASV_000079 | IV | Alteromonadales | 5.59 |
| ASV_000154 | IV | *Lutimonas* | 5.52 |
| ASV_000421 | IV | *Lutimonas* | 5.10 |
| ASV_000107 | IV | Bacteroidales | 4.77 |
| ASV_000039 | IV | Bacteroidales | 4.67 |
| ASV_000114 | II | *Desulfococcus* | 4.64 |
| ASV_000053 | IV | *Lutimonas* | 4.56 |
| ASV_000003 | IV | Alteromonadales | 4.51 |
| ASV_000175 | IV | Desulfobulbaceae | 4.29 |
| ASV_000292 | IV | *Lutimonas* | 4.18 |

Using the random forest approach (RF), all features were assigned a variable importance measure. For Norway and Scotland, the maximum variable importance was 13.3 and 14.7, respectively (Figs. 2A and 2B). For the top 20 ASVs ranked according to RF variable importance, the minimum values were 4.5 for Norway and 4.2 for Scotland.

The ASV variable importance measure of the RF approach revealed that among the 20 most important ASVs identified for the RF prediction model for Norwegian salmon farms,

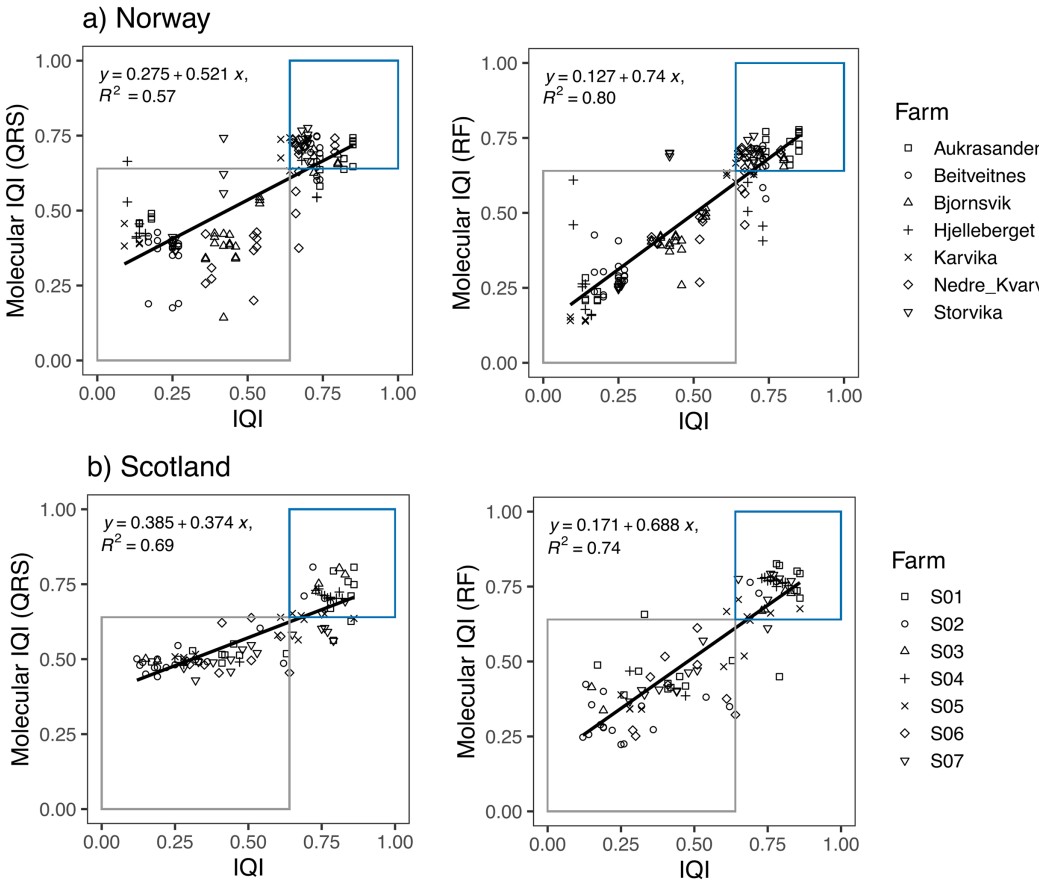

**Figure 3 Linear regression plots showing the relationship between the infaunal quality index (IQI) and the molecular IQI as estimated by quantile regression splines (QRS) and random forest (RF) for (A) Norway and (B) Scotland salmon farms.** The boxes indicate the two environmental quality categories that IQI assigns the samples (*i.e.*, blue for very good to good environmental quality samples and gray for moderate to poor environmental quality samples). Samples that are found inside the boxes are samples accurately predicted by the molecular IQI. The regression equation and the corresponding $R^2$ values are given for each regression plot.

15 were congruent with the good quality spline ASV indicators identified *via* QRS (Fig. 2A). These 15 indicator ASVs that were simultaneously identified by both approaches were taxonomically assigned to Gammaproteobacteria in Eco-Group II as assigned by QRS (sensitive taxa), Helicobacteraceae within the Eco-Group IV (transitory taxa), Flavobacteriaceae (Eco-Group IV) and other two ASVs belonged to the genus *Nitrospina* (Eco-Group II) and *Desulfosarcina* (Eco-Group IV) (Table 1A). Similarly, 15 out of the 20 most important ASVs identified by the RF model for the Scottish salmon farms had also good quality splines (Fig. 2B). These could be taxonomically assigned to the genus *Lutimonas* (categorized in Eco-Group IV), to the family Helicobacteriaceae (Eco-Group IV) and to *Psychrilyobacter* (Eco-Group IV) (Table 1B). Interestingly, only one single ASV (ASV_000107) was among the 20 most important variables in both RF models for Norwegian and Scottish salmon farms. In case of the Scottish salmon farms, this ASV was also identified as a good quality spline ASV. Taxonomically, this ASV could be assigned to Bacteroidales (Eco-Group IV).
**Table 2 Accuracy of quantile regression splines (QRS) and random forest (RF) predictions.**

| Method | % Accurate predictions | Adjusted $R^2$ | k |
|---|---|---|---|
| (A) Norway | | | |
| QRS | 89.9 | 0.57 (***) | 0.80 (***) |
| RF | 87.7 | 0.80 (***) | 0.75 (***) |
| (B) Scotland | | | |
| QRS | 88 | 0.69 (***) | 0.75 (***) |
| RF | 92.4 | 0.74 (***) | 0.84 (***) |

**Note:**

Percent of accurate predictions, regression coefficient for the relationship between the infaunal quality index (IQI) and the molecular IQI and Cohen's kappa statistic for the agreement of the IQI categories (very good to good or moderate to poor environmental quality) to the predicted IQI categories by QRS and RF for (A) Norway and (B) Scotland salmon farms. ***$p < 0.001$.

### Accuracies of molecular biotic index inferred with RF (*mol*-IQI$_{RF}$) and QRS (*mol*-IQI$_{QRS}$) compared to traditional IQI$_{MA}$

Linear regression models showed a high correspondence between the traditional IQI$_{MA}$ sample classification and both, the *mol*-IQI$_{QRS}$ and the *mol*-IQI$_{RF}$ for the Norwegian and also for the Scottish salmon farm dataset (Fig. 3). For both datasets, RF predictions had a higher coefficient of determination $R^2$ compared to QRS. In case of the Norwegian salmon farms, the discrepancy in $R^2$ was notably higher (0.8 for *mol*-IQI$_{RF}$ *vs* 0.57 *mol*-IQI$_{QRS}$) compared to Scottish salmon farms (0.74 for *mol*-IQI$_{RF}$ *vs* 0.69 *mol*-IQI$_{QRS}$). Interestingly, while the correlation coefficient $R^2$ of the *mol*-IQI$_{RF}$/traditional IQI$_{MA}$ was higher for the Norwegian than for the Scottish salmon farm samples (0.8 *vs* 0.74, respectively), the contrary was the case for the correlation coefficient $R^2$ of the *mol*-IQI$_{QRS}$/traditional IQI$_{MA}$.

The number of samples that were accurately predicted (*mol*-IQIs *vs* traditional IQI$_{MA}$) were in the same order of magnitude when comparing RF and QRS with each other as well as in a comparison of the two different geographic regions. For Norway ($n = 138$ samples), 89.9% of all samples were accurately predicted with *mol*-IQI$_{QRS}$ and 87.7% with *mol*-IQI$_{RF}$. For Scotland ($n = 92$ samples), *mol*-IQI$_{QRS}$ predicted 88% and *mol*-IQI$_{RF}$ 92.4%. Cohen's kappa statistics showed a significant ($p < 0.001$) agreement between both *mol*-IQI predicted classifications and the observed macrofaunal-inferred IQI$_{MA}$ classifications (good/moderate boundary) in both geographic regions. With kappa values of ≥0.8, the agreement between predictions and observations can be considered as "almost perfect" (k) for *mol*-IQI$_{QRS}$ predictions of Norwegian salmon farm samples and for *mol*-IQI$_{RF}$ predictions for Scottish salmon farm samples (Table 2). Kappa values for *mol*-IQI$_{QRS}$ predictions of Scottish salmon farm samples and for *mol*-IQI$_{RF}$ predictions for Norwegian salmon farm samples were still high (0.75 in both cases) but did not reach "perfect agreement".

### Erroneous classification obtained by *mol*-IQI$_{QRS}$ and *mol*-IQI$_{RF}$

Of the Norwegian salmon farm samples ($n = 138$), only 14 were classified erroneously with *mol*-IQI$_{QRS}$ and 17 with *mol*-IQI$_{RF}$ when considering the macrofauna-inferred IQI$_{MA}$ as

## a) Norway

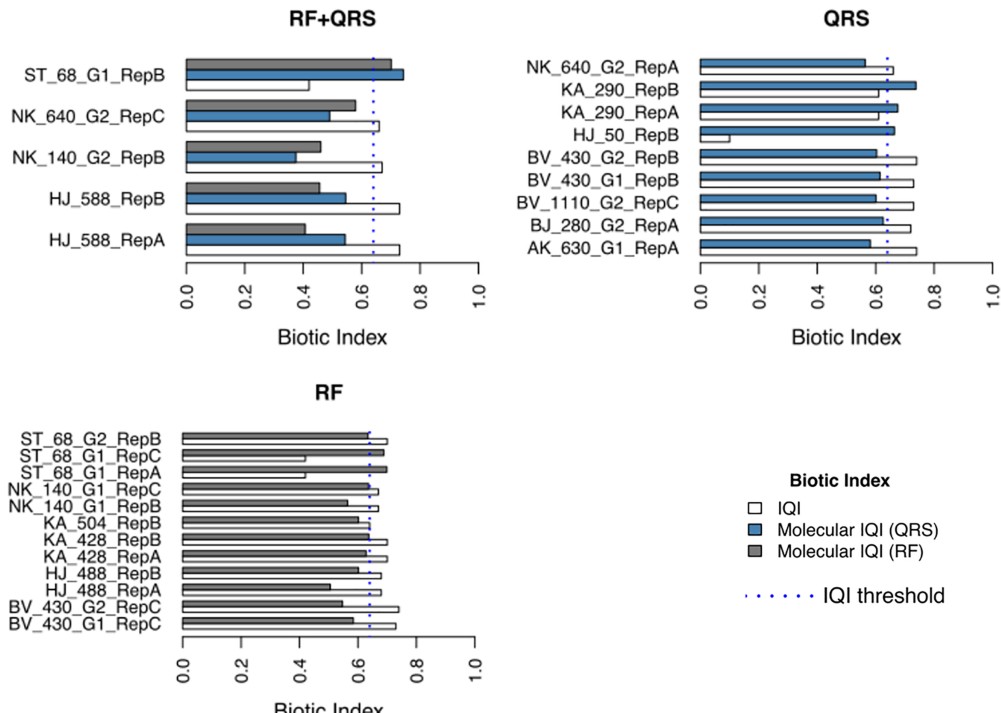

## b) Scotland

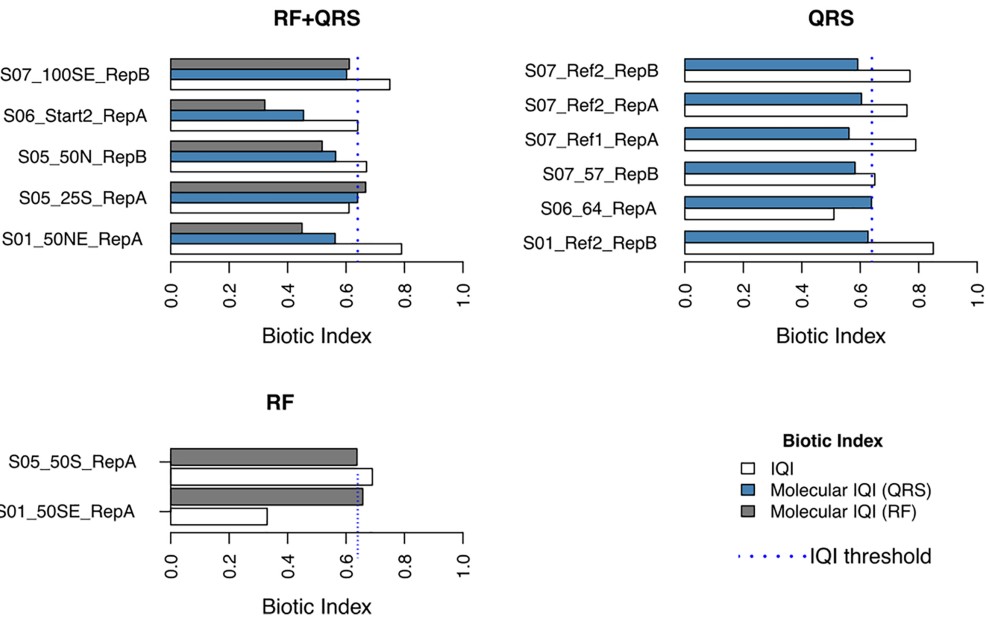

**Figure 4** **Erroneously predicted samples by quantile regression splines (QRS), random forest (RF) and both methods (RF+QRS) for (A) Norway and (B) Scotland salmon farms.** The vertical dotted line corresponds to the IQI threshold set to the 0.64 $IQI_{MA}$ good/moderate threshold.

ground truth (reference) (Fig. 4A and Table S3A). It is noteworthy that most of the erroneously classified samples (85%) had a reference $IQI_{MA}$ that differed ≤0.1 from the 0.64 good/moderate threshold, and, thus, were close to the classification decision boundary. In case of the erroneous $mol$-$IQI_{QRS}$ classifications these were 12 out of 14 samples that were close to the classification decision boundary. Interestingly, 10 of these 12 "close-to-classification-boundary" samples underestimated the reference $IQI_{MA}$ and classified these 10 samples into the "moderate to very poor" environmental quality class, while the actual reference $IQI_{MA}$ testified these samples a "very good to good" environmental quality ($IQI_{MA} > 0.64$). This observed pattern was the same for the $mol$-$IQI_{RF}$ classifications of Norwegian salmon farm samples. Fourteen out of 17 falsely classified samples were close to (≤0.1 deviation) the 0.64 $IQI_{MA}$ good/moderate threshold. All these 14 samples were underestimated with $mol$-$IQI_{RF}$ compared to the reference $IQI_{MA}$ and classified into the "moderate to very poor" environmental quality category while the reference $IQI_{MA}$ placed these samples into the "very good to good" class. In contrast, all erroneously classified samples from Norwegian salmon farms, which had a reference $IQI_{MA}$ that differed for >0.1 from the of 0.64 good/moderate classification boundary were overestimated by both $mol$-IQIs and falsely placed into the "very good to good" class (Table S3A). Thus, a clear pattern is obvious regarding over- and underestimation of reference $IQI_{MA}$ values using $mol$-$IQI_{QRS}$ and $mol$-$IQI_{RF}$-based environmental quality inference. This pattern is as follows: the vast majority of all false classifications by both molecular IQI indices referred to samples that were classified as "very good to good" environmental quality using the reference macrofauna-based $IQI_{MA}$. Both $mol$-IQIs placed these samples into the "moderate to very poor" environmental quality category (Fig. 4A).

A similar pattern could be observed for the samples from Scottish salmon farms. Most of the false classifications by both molecular IQIs (14 out of 18) underestimated the reference $IQI_{MA}$ and placed these samples erroneously into the "moderate to very poor" class rather than the "very good to good" class (Fig. 4B, and Table S3B). However, in contrast to the Norwegian salmon farm dataset, most falsely classified samples exhibited >0.1 difference from the 0.64 $IQI_{MA}$ good/moderate classification threshold. In detail, 11 of the 92 Scottish salmon farm samples were falsely classified with $mol$-$IQI_{QRS}$ and seven with $mol$-$IQI_{RF}$ (Fig. 4B and Table S3B). Only four of the eleven false $mol$-$IQI_{QRS}$ classifications of Scottish salmon farm samples corresponded to a reference $IQI_{MA}$ that was close to the 0.64 classification decision boundary (differed ≤0.1 from the 0.64 threshold). In case of the $mol$-$IQI_{RF}$ classifications, only four out of seven were close to the 0.64 $IQI_{MA}$ good/ moderate boundary. Once more three of these underestimated the reference $IQI_{MA}$ resulting in a false classification into the "moderate to very poor" category.

## DISCUSSION

Using quantile regression splines (QRS) and supervised machine learning (SML), we could infer the environmental quality (EQ) of 230 marine coastal sites subjected to organic enrichment due to aquaculture activities in two geographical regions with high accuracy (89% and 90%, respectively). Towards the development of a standard operating procedure

(SOP) for compliance monitoring of aquaculture impact on marine coastal environments, we investigated whether the traditional macrofauna-based Infaunal Quality Index ($IQI_{MA}$) shows a strong relationship with either or both of the molecular versions of the IQI that we inferred here with QRS (*mol*-$IQI_{QRS}$) and random forest (RF) algorithm (*mol*-$IQI_{RF}$) based on metabarcoding data. Our results showed that *mol*-$IQI_{RF}$ performed better than *mol*-$IQI_{QRS}$ in terms of correspondence with the macrofauna-based index, displaying a higher coefficient of determination ($R^2 = 0.8$ for Norwegian salmon farm samples, $R^2 = 0.74$ for Scottish salmon farm samples).

## ASV patterns along the organic enrichment gradient

QRS and RF showed good agreement in terms of important ASVs for RF and QRS-inferred bioindicators, which was reported before for the two methods (*Lanzén et al., 2021*). Among the 20 ASVs with the highest RF variable importance, 15 were also inferred as bioindicators for both Norwegian and Scottish salmon farms by QRS. On the other hand, the observed dissimilarities can be attributed to the fact that QRS is a method that analyzes each ASV individually and requires a consistent abundance response along the organic enrichment gradient as a prerequisite to consider them as bioindicators (*Keeley, Wood & Pochon, 2018*). In contrast, within the RF model, abundance information of individual ASVs can be processed not only individually, but also in combination with other ASVs to eventually measure the importance of ASVs (features) for the prediction of the biotic index (*Cordier et al., 2017*). From the top 15 ASVs identified as QRS-bioindicators with high RF variable importance (Table 1), we discuss here the ones that reached the genus identification level, *i.e.*, *Lutimonas*, *Psychrilyobacter*, *Nitrospina*, *Desulfosarcina* and *Desulfococcus*, while we refrain from further ecological interpretations which are not meaningful in higher taxonomic levels.

    *Lutimonas* (Flavobacteriaceae), which showed the highest RF variable importance, was assigned to Eco-Group IV, indicating poor ecological status. Representatives of Flavobacteriaceae have been previously reported in the salmon gut microbiome (*Fogarty et al., 2019*). Also, a *Lutimonas* species has been previously isolated from a marine polychaete (*Yang, Choo & Cho, 2007*), a taxonomic group (Annelida) traditionally used for bioindication of marine pollution (*Dean, 2008*). This is a promising result since it shows a potential connection between metabarcoding ASVs and the traditional marine bioindicators, as revealed also in other studies (*Pawlowski et al., 2014*; *Keeley, Wood & Pochon, 2018*). *Psychrilyobacter* was another bioindicator of poor ecological status within Eco-Group IV that was associated with high enrichment levels. *Psychrilyobacter* is a marine member of Fusobacteria isolated from marine sediment and has been characterized as an important degrader of the protein component in organic matter (*Zhao, Manno & Hawari, 2009*; *Yadav et al., 2021*). Even though, according to our results, *Lutimonas* and *Psychrilyobacter* have the potential to be used as bioindicators of poor environmental quality due to organic enrichment, their response should be further evaluated in new geographic regions and time periods to verify their bioindication power. To this end, targeted assays of quantitative PCR or digital PCR using ASV-specific primers would allow a fast screening of the presence and abundance of these ASV bioindicators in

environmental samples and their inclusion in routine application in biomonitoring (*Frühe et al., 2021a*).

Opposite patterns compared to literature were observed in the case of *Nitrospina, Desulfosarcina* and *Desulfococcus*. In our study, these taxa were associated with good ecological status (Eco-Group II), while in other studies they have often been reported at sites of bad environmental status, with *Nitrospina* being a known nitrite oxidizing bacterium (NOB) (*Lücker & Daims, 2013*) and *Desulfosarcina* and *Desulfococcus* being members of sulfate reducing bacteria (SRB) (*Kniemeyer et al., 2007*). This opposite observed pattern could be explained by the fact that co-influencing factors can eventually determine the structure of a bacterial community (*Frühe et al., 2021a*). For example, it is known that bacterial communities can be affected by seasonal changes in the environment, such as of temperature and nutrients (*Prosser et al., 2007*; *Gilbert et al., 2009*), but also by specific characteristics and conditions of the site such as the depth of the seafloor and/or the existing flow regime (*Cordier et al., 2018*). Additionally, short-term environmental alterations in our good ecological status sites might be reflected by the NOB and SRB patterns due to their fast response to environmental changes, but not yet by macrofauna patterns as suggested in previous studies (*Aylagas et al., 2021*). We would therefore suggest further collection and isolation of these bacteria under the fish cages and at reference sites (*Dowle et al., 2015*) along with the collection of more background environmental information (*Aylagas et al., 2021*).

## Performance of the QRS- and RF-inferred molecular biotic index

When comparing *mol*-IQI$_{QRS}$ and *mol*-IQI$_{RF}$ in terms of accuracy in the EQ classification, their performance can be considered equally good, since they both reached to "almost perfect" agreement with the IQI$_{MA}$ according to Cohen's kappa statistics. In both approaches there were erroneous classifications close to the 0.64 good/moderate IQI threshold, with 85% of the classifications in Norwegian salmon farms differing ≤0.1 from this boundary. This kind of misclassification leads to a borderline decision regarding the quality of the investigated samples. To improve classification power around the threshold, it is essential to collect more near-boundary samples which would improve group separability for QRS and RF. This would result in a better RF prediction performance near the threshold, as well as an increase in bioindicators identified *via* the QRS approach, enhancing the accuracy of the *mol*-IQIs.

Even though the EQ classifications were highly accurate for both QRS and RF approaches, an important parameter for their incorporation in biomonitoring programs would be that they exhibit high correlation with the traditional biomonitoring method, while it is supported that $R^2$ values below 0.8 could bring undesirable uncertainty that could lead to false assessments (*Keeley, Wood & Pochon, 2018*). Linear regression analysis did not show a strong correspondence between the macrofauna-based Infaunal Quality Index (IQI$_{MA}$) and the *mol*-IQI$_{QRS}$ ($R^2$ = 0.69 for Scottish salmon farm samples, $R^2$ = 0.57 for Norwegian salmon farm samples). In contrast to our findings, *Keeley, Wood & Pochon (2018)* found very strong relationships ($R^2$ = 0.9) between the traditional biotic index and the biotic index inferred using QRS. Their analysis resulted in a very high percentage of

operational taxonomic units (92%) with bioindication power that took part in the calculation of the biotic index. Here, from the total number of investigated ASVs, 60% were identified as bioindicators for Norwegian salmon farms and even a lower percent (30%) accounted for Scottish salmon farms. It is therefore possible that the low participation of bioindicators in the biotic index led to a decreased performance of the $mol$-IQI$_{QRS}$ which was even more evident for Scottish salmon farm samples (Fig. 3). The low number of organic enrichment specific bioindicators in Scottish salmon farms might be attributed to the fact that the collection sites were characterized by more shallow waters and thus compared to Norway there were more influences by other environmental parameters besides organic enrichment. In addition, the extracted QRS-inferred bioindicators in Scottish salmon farms were not evenly distributed into the eco-groups, with most good quality spline ASVs belonging to Eco-Group IV (poor ecological status), while the rest of the eco-groups were underrepresented, although the distribution of the input data was relatively even. *Frühe et al. (2021b)* suggested that more stable benthic microbial communities can be found close to farm cages since they are mainly influenced by the organic enrichment. This could potentially explain why most of the good quality splines in Scottish samples indicated poor ecological status, as Scottish sediment at sites less disturbed by organic enrichment might have been highly heterogeneous due to co-influencing environmental factors. In contrast to our results, *Keeley, Wood & Pochon (2018)* that achieved $R^2 = 0.9$ of linear regression between the traditional biotic index and the biotic index inferred using QRS, reported a more balanced distribution of their QRS-inferred bioindicators into eco-groups. The uneven representation of eco-groups has been discussed before as the major possible reason for the suboptimal performance of taxonomy-free ASV approaches and the inclusion of a larger sample pool in the future studies is suggested (*Frühe et al., 2021a*; *Apothéloz-Perret-Gentil et al., 2017*). Also, minimizing the seasonal effects, *e.g.*, samples collected in the same month and same depth as in *Keeley, Wood & Pochon (2018)* is expected to increase the number of bioindicators that get extracted from metabarcoding datasets, but also to improve the balance of indicators' proportions in EG.

The RF approach overall displayed better performance than QRS for both investigated regions, showing superior linear correspondence of the inferred *vs* actual values, as reported also in a previous study by *Lanzén et al. (2021)*. One possible reason that RF has an advantage over QRS is the fact that it is capable of handling samples containing a high amount of natural variability, which is a typical feature in metabarcoding datasets (*Frühe et al., 2021a*). This is because the RF algorithm itself is based on the technical method of bagging, also referred to as bootstrap aggregation, which induces variance reduction (*Breiman, 2001*). Similarly to our results, previous studies on Norwegian salmon farms repeatedly reported strong linear correlations (up to $R^2 = 0.85$ with kappa values close to 0.9) between SML-based ecological indices and macrofauna reference data (*Cordier et al., 2018*; *Cordier et al., 2019*). However, limitations of the SML-based approach were also discussed previously, with the main issues being incorrect references (*Frühe et al., 2021a*; *Cordier et al., 2021*) and inadequate sample coverage in the training dataset (*Gerhard & Gunsch, 2019*). In some cases, it was detected that the addition of samples without a reliable

macrofauna reference was responsible for the incorrect EQ assessment (*Frühe et al., 2021a*). Further, it was shown that for complex coastal sites, which are influenced by multiple stressors, a high number of samples is required to cover the dynamic community composition (*Prosser et al., 2007*; *Lanzén et al., 2021*; *Frühe et al., 2021a*). Additional diversification of the community is triggered by varying environmental parameters of different geographical regions and seasons, such as temperature, sampling depth, pH, redox potential, flow regimes and sediment types which introduce spatio-temporal heterogeneity of the samples (*Keeley, Wood & Pochon, 2018*; *Frühe et al., 2021b*; *Prodinger et al., 2021*).

To improve the accuracy of the SML-based model prediction, the collection of samples covering large spatiotemporal heterogeneity together with the monitoring of the environmental parameters in the sampling sites is needed. This allows the SML algorithms to disentangle background noise due to various environmental factors from the response to our target environmental stressor, such as the organic enrichment (*Frühe et al., 2021b*). For example, information such as season, depth or temperature can be co-learned when added as additional features to the SML. Consequently, the algorithm would automatically determine a season-, depth-, or temperature specific set of ASVs which can be used for the prediction of a new sample influenced by similar environmental conditions. In the future, a universally applicable monitoring tool can be established using a big variety of samples in the SML models which will then detect more robust indicator ASVs interchangeable across biogeographic regions (*Frühe et al., 2021a*).

Finally, it should be noted that the number of required sequences, and thus the number of features/ASVs needed for a benchmark prediction, varies per dataset (*Dully et al., 2021a*). To get an estimate on how many features are needed for an accurate prediction, *Dully et al. (2021a)* proposed to create an ordination analysis, *e.g.*, non-metric multidimensional scaling which can potentially display the degree of class separability. If the class separability is good, the algorithm needs less features to achieve a correct prediction. For this study, in order to obtain a direct comparability between the SML and QRS method, we selected only the 250 most abundant ASVs to be used as features for the SML prediction and we investigated their representation in our datasets (each ASV contributing ≥0.04% to the total number of reads), as well as that the variation included in the dataset can be well represented by those 250 ASVs (Fig. S4). However, it should be noted that for the development of a universally applicable monitoring system, all available ASVs should be incorporated into the SML models in order to have numerous bioindicator candidates that are recurring globally with a consistent response to the targeted environmental factor.

## CONCLUSIONS

In conclusion, even though both approaches are promising to infer environmental quality based on metabarcoding data, SML is capable of handling the natural variability which can determine bacterial responses to environmental stressors (*Aylagas et al., 2021*). For QRS approach, more research on the response of the indicator ASVs to organic enrichment and the co-influence of other environmental parameters is necessary to eventually select the

most powerful stressor-specific indicators. Overall, we recommend that efforts should be focused on the improvement of the SML approach, so that it could be efficiently applied for environmental quality assessments at marine ecosystems subjected to organic enrichment. This would be achieved with the addition of new samples in the model along with additional measurements of environmental parameters (*Cordier, 2020*; *Lanzén et al., 2021*). As additional samples are added, the algorithm learns to distinguish natural variations in the bacterial communities from variations introduced by organic enrichment (*Frühe et al., 2021a*) and stressor-specific ASVs can be detected. Additionally, we encourage scientists to measure environmental parameters while sampling, because these can be easily incorporated into the algorithm as additional features (*Cordier, 2020*). This will enable future studies to disentangle potential background noise introduced by spatiotemporal heterogeneity from the response to the target stressor, *i.e.*, organic enrichment, thus improving the predictive power (*Cordier et al., 2017*; *Cordier, 2020*). Finally, for the successful implementation of the SML approach, the number of features that provide enough information should be investigated. To get an estimate on how many features are needed for an accurate prediction it is proposed to create an ordination analysis, *e.g.*, a non-metric multidimensional scaling as it can potentially display the degree of class separability (*Dully et al., 2021a*). For the development of a universally applicable monitoring system all available ASVs should be incorporated into the SML models in order to have numerous bioindicator candidates that are recurring globally with a consistent response to the targeted environmental factor.

## ACKNOWLEDGEMENTS

We are grateful to Mowi Norway, Mowi Scotland and Scottish Sea Farms for providing samples or access to salmon farms. We also thank Iain Berrill of Scottish Salmon for his continuous and valuable support of our research to improve biomonitoring in the context of salmon aquaculture.

### Funding

This study received financial support from the Deutsche Forschungsgemeinschaft (STO414/15-2). The funders had no role in study design, data collection and analysis, decision to publish, or preparation of the manuscript.

### Grant Disclosures

The following grant information was disclosed by the authors:
Deutsche Forschungsgemeinschaft: STO414/15-2.

### Competing Interests

The authors declare that they have no competing interests.

## Author Contributions

- Kleopatra Leontidou performed the experiments, analyzed the data, prepared figures and/or tables, authored or reviewed drafts of the article, and approved the final draft.
- Verena Rubel performed the experiments, analyzed the data, prepared figures and/or tables, authored or reviewed drafts of the article, and approved the final draft.
- Thorsten Stoeck conceived and designed the experiments, performed the experiments, authored or reviewed drafts of the article, and approved the final draft.

## DNA Deposition

The following information was supplied regarding the deposition of DNA sequences:

The raw DNA sequences are available at SRA BioProject: PRJNA947566.

https://www.ncbi.nlm.nih.gov/bioproject/PRJNA947566

## Data Availability

The data and R code are available at figshare: Leontidou, Kleopatra; Rubel, Verena; Stoeck, Thorsten (2023): Script from "Comparing Quantile Regression Spline analyses and Supervised Machine Learning for environmental quality assessment at coastal marine aquaculture installations". figshare. Dataset. https://doi.org/10.6084/m9.figshare.21077857.v1

## Supplemental Information

Supplemental information for this article can be found online at http://dx.doi.org/10.7717/peerj.15425#supplemental-information.

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
