# Peer review of "Comparing quantile regression spline analyses and supervised machine learning for environmental quality assessment at coastal marine aquaculture installations"

_PeerJ, doi:10.7717/peerj.15425_

## Round 0.1 · original submission · Minor Revisions

Both reviewers appreciated this paper, and have provided a few minor suggestions that I think will improve the manuscript.

Reviewer 1 ·

Basic reporting

no comment

Experimental design

To ensure a meaningful and robust dataset, the authors combined previously published and newly generated sequencing data obtained from a well-designed sampling plan. With regards to the actual sampling sites, I was missing some metadata information (location, depth, distance from cage edge and any environmental parameters measured) from the submission and could not find such data in the figshare repository either (coordinates are included in the SRA database though).

The authors do refer to such information being available in Table S1:
lines 186-187: “In brief, sediment was collected at 5-10 stations (depending on farms, see Table S1)”

However, Table S1 does not contain information about the sampling sites – it shows summarized sequencing data for the 5 Scottish farms.

1. Please include a metadata table which provides the available details regarding all sampling sites (including those from previously published data)
2. It would be useful as well to show the sequencing statistics per sampling site (possibly in the same metadata table)
3. Finally, a map indicating the location of the different farms and stations would be excellent

Validity of the findings

In their discussion, the authors elaborated on the potential biases in the dataset that could have influenced the observed outcomes and suggested ways to counteract them in future studies.

In this regard, I would like to highlight one point:
lines 539-541: “To improve classification power around the threshold, it is essential to collect more near-boundary samples which would improve group separability for QRS and RF.”

Here, the authors touch upon one important aspect, namely a potential class imbalance of the collected samples. Although it seems that there was a good spread of IQI values for both the Norwegian and Scottish sites, it could be useful to indicate the distribution of IQI classes for the samples included in this method comparison.

Additional comments

I would like to praise the authors for submitting a text that is very clear and unambiguous throughout all sections. In particular, the two approaches (QRS and SML) are very clearly explained in the Introduction section highlighting the pros and cons of each.

·

Basic reporting

The manuscript by Leontidou et al. describes the comparative performance of Quantile Regression Splines and Supervised Machine Learning in the scope of environmental monitoring, specifically using these methods to elucidate ecosystem impacts of aquaculture operations. These higher-throughput methods could replace the traditional used characterization of benthic macroinvertebrate bioindicators which is cumbersome and requires major effort to complete.

The manuscript is well written, has proper English and is scientifically sound. I appreciate the clear introduction which sets out some key elements, and references the relevant literature.

While the article is structured well and has all the relevant sections, I do have to comment on the figure and table legends, which I feel are not of the same quality as the rest of the data that is being presented. The legends of figure 1 and 2 for example are quite terse and do not fully explain the figure as a standalone object. Also, insets are not always clearly described (like in figure 2, A and B is missing).

Experimental design

The experimental design is interesting and solid. I have only small comments on the technical aspects of this paper.

It is unclear to me why an arbitrary value of 250 was picked to select ASV's to follow up on (line 249). As stated later in the text, this amounts to only a tiny fraction of the complete diversity in these samples, and I feel this might obscure a lot of trends, especially in the SML approach. Even low abundance ASVs can translate to major ecosystem function or appear as co-occurring entities, and trimming the data to just to top 250 ASV's seems a strange decision to me. More so since one of the study goals is to create a scalable and automatable workflow, in which the "load" so to speak should not matter. I am not expecting the authors to redo all the analysis, but the decision process to arrive at 250 should at least be discussed and perhaps ways to overcome this limitation should be explored.

Another comment I have is that this paper uses data from previous work and newly generated data. I appreciate that the data from the previous study does not necessarily need to be repeated in depth, but in my opinion it would be nice to see all the data represented in the rarefaction curves (Fig S1), and the sequence data table (Table S1) so readers can compare if they want to.

Additionally, since the data is from 2 different sequencing runs, and data from these 2 runs is later combined into data-sets, I wonder if the authors should at least mention the possibility of batch-effects?

Validity of the findings

To the best of my knowledge, all underlying data has been provided and the findings are valid.

I did miss a section which summarizes the results for the SML section in this paper, for example there is a section dedicated to QRS ("Eco-Group assignment and identification of potential bacterial ASV indicators using Quantile Regression Splines (QRS)") but no similar result section is found for SML.

In addition, I was hoping to see some cross validation between Scotland and Norway. Are the models and ASV types interchangeable? Will it be possible in the future to create a unified SML model, or a set of ASVs that can be easily deployed over heterogeneous environments?

Additional comments

Some small specific comments, please address them as you see fit (line numbers included):

126: Why will this remain unanswered? Unclear.

128: I am not sure what a "reasonable" amount of samples is, and that could be different depending on the situation. I would formulate this differently.

215 and 219: Vendor NEB is used but not defined

246: What was the range of # of sequences, was there a very large difference between them? Maybe indicate min/max seq per sample in table S1.

249: As discussed above, it is unclear to me why an arbitrary value of 250 was picked for the molecular IQI inference

261: This sentence runs a bit strange, maybe “for each of the”

292: Equation 1 is referenced but not defined, but it may be that the heading for equation 1 will be added during production? In that case, please ignore this comment.

350: It would be nice to have the information of all farms used in this study consolidated in table S1 and figure S1.

360: Figure 1 is referenced but it has 2 panels (A and B), please add in text.

395: Figure 4 is referenced, but this figure shows erroneous predicted samples? It seems this reference is not correct?

529: Missing closing parentheses

531: Should the difference between OTU’s and ASV’s be highlighted? Could this be a factor in the difference observed?

Sup. Fig S1: Maybe some meta-data could be coded here in color, like sequencing run A/B, or Norway/Scotland? To have a high level overview on how the sequencing performed in terms of output?

Sup. Fig S2: “The observed peak values of ASV relative abundance within a farm are indicated with vertical lines.”. At first I did not understand, since it is reference the axis on which the IQI is displayed, but I assume it is meant the IQI value at which the ASV has its peak abundance? Maybe this could be better worded for clarity.

Reviewer 3 ·

Basic reporting

See attachment.

Experimental design

See attachment.

Validity of the findings

See attachment.

Annotated reviews are not available for download in order to protect the identity of reviewers who chose to remain anonymous.

---

## Round 0.2 · accepted · Accept

Thanks for addressing all of the previous comments and congratulations on the publication!